# Dietary Changes and Anxiety during the Coronavirus Pandemic: Differences between the Sexes

**DOI:** 10.3390/nu13124193

**Published:** 2021-11-23

**Authors:** Mona Boaz, Daniela Abigail Navarro, Olga Raz, Vered Kaufman-Shriqui

**Affiliations:** 1Department of Nutrition Sciences, Ariel University, Ariel 40700, Israel; abigaily@ariel.ac.il (D.A.N.); olgara@ariel.ac.il (O.R.); veredks@ariel.ac.il (V.K.-S.); 2Centre for Urban Health Solutions (C-UHS), St. Michael’s Hospital, 209 Victoria St., Toronto, ON M5B 1W8, Canada

**Keywords:** COVID-19, Mediterranean Diet, anxiety, dietary patterns, sex

## Abstract

The SARS-CoV-2 (COVID-19) pandemic has been associated with both increased anxiety, deterioration in diet and weight gain. These associations may differ by sex. The present report examines differences by sex in diet quality in order to determine whether associations between diet and psychological stress during the initial phase of the COVID-19 pandemic differed by sex. This online study is available internationally in seven languages. The Mediterranean Diet Score was used to measure diet quality, while the General Anxiety Disorder 7-point scale (GAD-7) was used to measure anxiety. Findings were compared by self-reported sex (male vs. female). A total of 3797 respondents provided informed consent and met eligibility criteria, of whom 526 women were omitted due to being pregnant or six months or less post-partum, or due to reproductive status not being reported. Thus, 3271 individuals are included in the present report, of whom 71.2% were women. The median age of women was 30 (interquartile range (IQR) = 16) years vs. 31 (IQR = 19) years, *p* = 0.079. The median diet quality score was 9 (IQ = 3) in both women and men (*p* = 0.75). Despite the overall similarity in diet score, several components of the score differed significantly by sex. Women reported consuming significantly more olive oil, daily servings of vegetables, and weekly servings of sweet baked goods. Men reported consuming significantly more sweetened/carbonated drinks, red meat, alcohol, legumes, and hummus/tahini. Women reported a GAD-7 score of 6 (IQR = 8), while men reported 3 (6), *p* < 0.001. An inverse association was detected between the Mediterranean diet score and the GAD-7 score in both women (rho = −0.166, *p* < 0.001) and men (rho = −0.154, *p* < 0.001), and the correlation coefficients did not differ by sex (*p* = 0.76). Mediterranean diet score and age both reduced the odds of elevated anxiety (GAD-7 ≥ 10), while female sex, deterioration of diet quality during the outbreak, unemployment, and completing the survey in English increased the odds of this outcome. During the COVID-19 lockdowns, overall diet quality did not differ by sex; however, some differences by sex in components of the total score were detected. Moderate to severe anxiety was positively associated with female sex and poorer diet quality even after controlling for age, employment status, and the language in which the survey was performed.

## 1. Introduction

Governments worldwide implemented public health strategies in response to the global coronavirus SARS-CoV-2 (COVID-19) pandemic, including mitigation and/or suppression to control the outbreak [1,2]. The pandemic itself as well as the measures implemented to control it have been associated with increased anxiety, possibly attributable to uncertainty and social isolation [3,4].

Anxiety has been associated with disruption of daily routines, including maintaining a healthy diet, exercise and even personal hygiene [5]. An increase in weight during the COVID-19 pandemic is related to decreased physical activity, increased “emotional eating” and baseline overweight/obesity level [6]. Though the direction of causality has not been fully elucidated, the association between anxiety and obesity is pertinent due to increased adverse COVID-19 outcomes in people with obesity [7,8]. The disease is characterized by vascular inflammation, endothelial dysfunction, and a hypercoagulation state [9], a pathogenesis potentially amplified in the pro-inflammatory metabolic environment typical of obesity [10]. Anxiety has been inversely associated with diet quality and positively associated with weight gain during the pandemic [11].

The Mediterranean diet (MedDiet) is associated with reduced mortality and is exemplified by regular intake of olive oil, fruits and vegetables, whole grains, pulses, fish, and nuts, together with infrequent consumption of added sugars and red and/or processed meat [12,13,14]. Thus, the MedDiet score can be used to approximate diet quality [15]. Importantly, MedDiet score has been shown to be inversely associated with psychological stress among adults [16].

Eating behaviors are known to differ by sex. For example, in a validation study in which the Adult Eating Behavior Questionnaire (AEBQ) was translated to Polish for use in adolescents participating in the Polish Adolescents’ COVID-19 Experience (PLACE-19) Study, girls reported greater food responsiveness, emotional over- and under-eating, ability to control food intake in response to satiety, and eating slowness than boys [17].

Prevalent anxiety has been shown to differ by sex, such that both lifetime prevalence and disease burden are greater in women than men [18]. On the other hand, COVID-19 mortality is greater in men than women in most countries [19].

With both eating behaviors and anxiety differing by sex, and COVID-19 mortality also differing by sex, the present study was designed to determine whether the association between diet quality and anxiety differed by sex during the COVID-19 pandemic.

## 2. Objectives

The present cross-sectional study was designed to determine whether sex influenced the association between diet quality and anxiety during the early phase of the COVID-19 pandemic.

## 3. Methods

### 3.1. Study Design

A cross-sectional survey was conducted online. A convenience sample of respondents was used to quickly maximize response rate during the early COVID-19 pandemic. The survey measured diet quality using Mediterranean diet (MedDiet) score. The General Anxiety Disorder 7 (GAD-7) score was used to measure anxiety. Correlations between these measures represented the degree to which diet and anxiety were associated during the outbreak. Respondents self-reported sex (male, female, other). The report aimed to assess the degree to which sex influenced the aforementioned associations.

### 3.2. Ethics

The study was approved by the Helsinki Committee (Institutional Ethics Board) of Ariel University, Israel (Approval number AU-HEA-VKS-20200329, 29 March 2020). Informed consent was provided by each participant prior to responding to the survey by indicating such on the survey. The study was registered at Clinicaltrials.gov (accessed on 16 December 2020), NCT04353934.

### 3.3. Study Location

Google Survey was the platform used to conduct this online survey. A link to the survey was uploaded to several social media sites, such as the Department of Nutrition, Ariel University Facebook pages in both Hebrew and English. The r/Coronavirus community on reddit created a page for pandemic-associated research and the survey link was uploaded there. Additionally, investigators and department staff uploaded the survey link to their own social media pages. Participants were urged to re-post the survey link to social media sites and pages, creating a sort of social media snowball recruitment.

### 3.4. Study Population

All adult (reported age 18 years or older) individuals who provided electronic informed consent and completed the survey online were included.

### 3.5. Inclusion Criteria

All non-pregnant, non-post partum adults who responded to the survey after providing informed consent in any of the study languages (Arabic, English, French, Hebrew, Italian, Spanish or Russian) were included in the study.

### 3.6. Exclusion Criteria

Subjects who failed to provide informed consent and those who indicated that their age was younger than 18 years were excluded from the analysis. In the present analysis, pregnant women or those indicating they had given birth within the last six months prior to participating in the survey, as well as those who did not indicate reproductive status, were omitted due to possible confounding with anxiety (*n* = 526, or 13.8% of analyzable cohort).

### 3.7. Study Procedures

A link to the survey was posted to social media pages. Interested individuals could respond to the survey after providing informed consent. Responses were recorded to Google Survey, which generates an Excel spreadsheet. This was uploaded to SPSS v25 for analysis.

### 3.8. Data Acquisition and Survey Characteristics

Elicited were: (1) demographic information including age, sex, employment status, education and profession; (2) the nutrition questionnaire (MedDiet); and (3) the GAD-7. Data were anonymous; however, respondents could provide an e-mail address to which the MedDiet score would be mailed upon request.

### 3.9. Survey Translation

The survey was developed in Hebrew. Expert validity and face validity were assessed by a panel of registered dietitians. Next, the survey was piloted among Hebrew speaking volunteers (students, administrative staff and their families) in order to ascertain readability, comprehension and reliability, as the survey was re-administered to a subset of pilot participants after a two-week interval. The survey was then translated from Hebrew to each of the six other languages: Arabic, English, French, Italian, Russian, and Spanish. Each translation was performed by a native speaker of the target language who was also fluent in Hebrew. Each translation was then back-translated to Hebrew by another individual who was bilingual in both the target language and Hebrew, thus validating the translation. The translated surveys were piloted in small groups of native speakers of each target language for readability and comprehension.

### 3.10. MedDiet Score

The Israeli Mediterranean diet screener (I-MEDAS) was used to measure diet quality defined as similarity of the diet to the MedDiet. To score the I-MEDAS, a point value of 1 is assigned to a given question if the criterion is met, and 0 if the criterion is not met. Thus, the total I-MEDAS score can range from 0 to 17 points. The total I-MEDAS score has been shown to have predictive utility for mortality such that each 1-point increase in score was related to a 12% relative reduction in risk of death [20]. The score was used to assess diet quality independent of cultural framework rather than to capture the dietary intake of study participants, who were from 79 different countries.

### 3.11. Anxiety Score

The seven-item self-reported Generalized Anxiety Disorder Scale (GAD-7) was used to assess anxiety. The GAD-7 has been validated for use in the general population in many languages, including each of the languages into which the survey was available [21]. The GAD-7 asks respondents to respond to a set of statements with reference to the two weeks prior to the survey. Each statement is scored as follows: 0 (not at all), 1 (several days), 2 (more than half of the days) 3 (nearly every day). Consequently, the total GAD-7 score can range from 0–21, where a higher score indicates greater anxiety [22]. Cut-off scores for mild, moderate, and severe anxiety symptoms are 5, 10, and 15, respectively [23,24].

### 3.12. Sampling Procedures

The survey was conducted using a convenience sample such that adults who provided informed consent and responded to survey questions could be included. A probability sample was not attempted. Respondents were encouraged to re-post the survey to their social media pages, thus adding a snowball distribution element to the sampling procedure.

### 3.13. Statistical Analysis

Data were downloaded from Google Survey to Excel (Microsoft, Redmond, WA, USA) and analyzed on SPSS v25 Statistical Analysis Software (IBM Inc., Armonk, NY, USA). Distributions of continuous variables were assessed for normality using the Kolmogorov–Smirnov test. Distributions of all continuous variables deviated significantly from normal, and thus are described as median, interquartile range (IQR). Nominal variables were presented as n (%). Associations between continuous variables were described by calculating the Spearman’s correlation coefficient. Continuous variables were compared by nominal variables using the Mann–Whitney U or Kruskal–Wallis tests as appropriate. Associations between nominal variables were assessed using the chi-square test. A logistic regression model was developed to identify moderate to severe anxiety (GAD-7 score ≥ 10) [21], into which variables by which GAD-7 differed in univariate analyses were included. The final model was arrived at using a backward approach with the likelihood ratio test, with an entry probability of 0.05 and a removal probability of 0.1. Odds ratios with 95% confidence intervals were calculated for each included covariate. All analyses were two-sided and considered significant at *p* < 0.05.

### 3.14. Sample Size and Study Power

The original sample size calculation indicated that 2144 participants would provide a 95% confidence level and a confidence limit of 2% for a GAD score of 10 or greater. This sample size also provided 99% power to detect a true association between the GAD score and MedDiet score of r = 0.15 or greater, with alpha = 0.0001. The sample in the current report (*n* = 3271) size exceeds the calculated sample size by 52%.

## 4. Results

### 4.1. Database and Participant Dispensation

On 25 April 22:00 Jerusalem time (GMT + 2), the questionnaire was closed to responses. Participant dispensation is depicted in Figure 1. Of the 4028 individuals who initiated the survey, 3941 completed and submitted it. Excluded were 87 respondents for whom informed consent was not provided, 49 duplicate responses, and eight respondents whose indicated age was younger than 18 years. A total of 3797 were included in the initial analysis. Of these, women who indicated that they were currently pregnant or had delivered during the six months prior to answering the survey (*n* = 117) and those who did not indicate reproductive status (*n* = 409) were omitted from the current analysis, leaving *n* = 3271 participants in the present analysis (*n* = 2328 women).

Respondents completed the survey in all languages into which it had been translated. A total of 347 participants indicated they were from European countries: Albania, Austria, Belgium, Croatia, the Czech Republic, Denmark, Germany, Faroe Islands, Finland, France, Greece, Hungary, Iceland, Ireland, Italy, Latvia, Malta, Moldova, North Macedonia, Norway, Poland, Portugal, Romania, Russia, Serbia Slovakia, Slovenia, Spain, Sweden, Switzerland, the Netherlands, the United Kingdom, and Ukraine.

North American participants included Canada (*n* = 135) and a very large number from the United States (*n* = 1140).

There was a total of 57 respondents from South and Central America, including Argentina, Brazil, Chile, Columbia, Costa Rica, El Salvador, Honduras, Mexico, Panama and Peru.

The largest number of participants was from Israel (*n* = 1897). Another 186 came from other Asian countries, including China, Georgia, India, Japan, Kazakhstan, Malaysia, Nepal, Pakistan, the Palestinian Authority, the Philippines, Saudi Arabia, Singapore, Sri Lanka, Thailand, Turkey, the United Arab Emirates and Viet Nam.

The six African participants came from five countries: Cameroon, Nigeria, South Africa, Tanzania, and Tunisia.

The final 29 participants came from Oceania and the Pacific Islands and included Australia, New Zealand and the Northern Mariana Islands.

### 4.2. Participant Characteristics

Table 1 presents study population characteristics by sex. Of the 3271 participants in the present survey, more than 71.2% were female. Most respondents answered the survey in English or Hebrew, with a by-sex difference in the distribution of languages in which the survey was completed was detected. Most respondents did not currently or previously have COVID-19, but 17 (0.5%) did report either a confirmed diagnosis at the time of completing the survey or recovery prior to the survey. Another 75 (2.3%) indicated their status as “suspected” infection without laboratory confirmation. COVID-19 status did not differ by sex. Significantly more men than women reported current smoking. Usual employment settings differed significantly by sex, with more women in public sector/nonprofit venues and more men in large/medium private industry or independently-owned businesses. Employment status during the pandemic lockdowns did not differ significantly by sex. Education also differed by sex, with a greater percentage of women reporting university education.

### 4.3. Lifestyle and Dietary Characteristics

Presented in Table 2 are participant lifestyle and dietary characteristics by sex. The median MedDiet score was 9 (3) in both men and women and did not differ by sex. Despite this overall similarity in diet score, by-sex differences in score components were detected. Specifically, women reported consuming significantly more olive oil, daily servings of vegetables, and weekly servings of sweet baked goods, while men reported consuming significantly more sweetened/carbonated drinks, red meat, alcohol, legumes, and hummus/tahini.

Relative to their pre-pandemic diets, most respondents, regardless of sex, indicated that their diets were healthier prior to the pandemic; however, significantly more women than men indicated a worsening of diet quality. By contrast, more than twice as many men than women considered their diet to have improved during the pandemic.

More than a quarter of the women and a fifth of the men reported increased weight since the pandemic onset, and the pattern of weight change (gain or loss) during this period differed by sex. Change in weight was associated with change in diet quality; specifically, both women (89.3%) and men (83.7%) who gained weight during the pandemic reported a deterioration in diet quality during this same period. Among respondents who reported weight gain during the outbreak, the median (IQR) MedDiet score in both women and men was 8(3). By comparison, the median (IQR) MedDiet score among those who reported no change in weight or weight loss during the pandemic was 9 (4) in both women and men, indicating a significant inverse association between MedDiet score and change in body weight (*p* < 0.001).

Significantly more women than men reported following a vegetarian or vegan diet. The median (IQR) MedDiet score in both women and men following these diets was 10 (3) vs. 9 (3) in non-vegetarians/vegans, *p* < 0.001. More women than men reported taking nutrition supplements. The median (IQR) MedDiet score did not differ by nutrition supplement use in women, at 9 (3) in both groups. In men, the median (IQR) MedDiet score was significantly higher in those who reported using nutrition supplements compared to those who did not: 9 (3) vs. 9 (4), *p* = 0.04.

A 50% relative reduction in the time devoted to exercise during the pandemic was detected, declining from a median of 120 to 60 min per week; this pattern was similarly observed in both women and men.

### 4.4. Anxiety

Measures of anxiety are displayed in Table 3. The total GAD-7 score was consistent with the cut-off for mild anxiety in women and was significantly higher than the GAD-7 score in men. Women also had significantly higher scores for each of the GAD-7 score components than men.

### 4.5. Associations between Anxiety and MedDiet Score

Table 4 presents correlations between the GAD-7 score and the MedDiet score as well as each of its component items.

An inverse association was detected between the MedDiet score and the GAD-7 score in both women (rho = −0.166, *p* < 0.001) and men (rho = −0.154, *p* < 0.001), and the correlation coefficients did not differ by sex (*p* = 0.76).

In women, the GAD-7 score was significantly, positively associated with the following items of the MedDiet score: daily servings of butter, margarine, or cream; sweetened beverages per day; whole-grain servings per day; servings of red or processed meat per week; alcoholic beverages per week; weekly servings of legumes; servings of savory baked goods per week; and servings of salty snacks per week. In men, items significantly, positively associated with the GAD-7 score included: servings of butter, margarine, or cream per day; weekly servings of red/processed meat; sweetened beverages per day; weekly servings of savory baked goods; and weekly servings of salty snacks.

Significant inverse associations were identified between each of the following MedDiet items and the GAD-7 score in women: use of olive oil as the main culinary fat; servings of vegetables per day; daily servings of unsweetened dairy products; weekly servings of fish; weekly servings of nuts; and servings of hummus/tahini per week. In men, significant inverse associations were observed between the following MedDiet items and the GAD-7 score: servings of fish per week; weekly servings of nuts; and weekly servings of hummus/tahini.

Associations between MedDiet components and GAD-7 that significantly differed by sex included daily servings of fruit, though fruit was not significantly associated with GAD-7 score in either sex; daily servings of butter/margarine/cream, which was significantly positively associated with anxiety in both sexes but more strongly in women than men; servings per day of unsweetened dairy products, which was inversely associated with anxiety in women and not significantly associated with anxiety in men; and weekly servings of alcoholic beverages, which was significantly positively associated with anxiety in women, but not significantly associated with anxiety in men.

Women who reported a healthier diet prior to the onset of the COVID-19 pandemic had a median (IQR) total GAD-7 score of 7 (9), compared to 4 (6) among those who reported no change in diet quality and 5 (6) among those reporting an improvement in diet quality since the start of the outbreak. In men, the median (IQR) GAD-7 values were 4 (6) among those who reported healthier diets prior to the pandemic, 1 (4) among those who reported no change, and 3.5 (7) among those reporting improved diet quality during the outbreak.

Change in body weight during the pandemic was also associated with anxiety. Median (IQR) GAD-7 score was 7 (10) in women who reported weight loss, and 7 (9) in women who reported weight gain. While these measures did not differ significantly from one another (*p* = 0.532), they both differed from women who reported no change in weight during the pandemic 5 (7), *p* < 0.001. In men, median (IQR) GAD-7 scores were 4 (10) in those who reported weight loss, and 5 (6) in those who reported weight gain (*p* = 0.98). Both values differed significantly from men who reported no change in body weight during the COVID-19 outbreak (*p* = 0.001). GAD-7 scores were significantly higher in women than in men in each category of weight change.

### 4.6. Anxiety and Respondent Characteristics

The GAD-7 score was significantly inversely associated with age in both women (rho = −0.122, *p* < 0.001) and in men (rho = −0.136, *p* < 0.001). Anxiety scores differed significantly across the languages in which the survey was performed as depicted in Figure 2. In women, respondents who completed the survey in English had the highest median (IQR) GAD-7 scores, 9 (10), while those who completed the survey in Hebrew reported the lowest median (IQR) GAD-7 scores, 3 (5). In men, the same pattern was observed, such that the highest median (IQR) GAD-7 scores were in those who completed the survey in English, 6 (9), and the lowest scores were in those who completed the survey in Hebrew, 1 (4). A significant sex-by-language interaction was detected (*p* = 0.018). Median (IQR) GAD-7 scores were highest among unemployed women 8 (10) and men 5 (9.75), and these values differed by sex, *p* < 0.001.

Mild anxiety (GAD-7 ≥ 5) was reported by 60.5% of women vs. 39.9% of men (*p* < 0.001), while moderate to severe anxiety (GAD-7 ≥ 10) was reported by 30.5% of women and 16.5% of men (*p* < 0.001). Presented in Table 5 is the multivariable logistic regression model of moderate to severe anxiety (a GAD-7 score of 10 or greater). Initially entered into the regression model were variables by which the GAD-7 score differed in univariate analysis, and included age, sex, the language in which the questionnaire was completed (English vs. any other language), change in diet pattern (diet healthier prior to COVID-19 yes/no), smoking status (current smoker yes/no), employment status (unemployed yes/no), minutes per week of physical activity during the pandemic, and MedDiet score. The final model was arrived at using a backward, stepwise approach. Physical activity and current smoking status were not included in the final model. The model was significant, and correctly classified 74.4% of study participants for moderate or more severe anxiety. As can be seen, women were more than twice as likely as men to experience moderate to severe anxiety. Individuals who completed the survey in English were more than four times as likely to have moderate to severe anxiety. Unemployment increased the odds of moderate to severe anxiety by 36%. People who reported that their current diets were less healthy than their diets prior to the COVID-19 pandemic had a 70% increase in odds of moderate to severe anxiety. On the other hand, each 1-point increase in the MedDiet score reduced the risk of moderate to severe anxiety by a relative 7.2% (4–11%). Age also significantly reduced the risk of moderate to severe anxiety.

## 5. Discussion

In the present study, women reported significantly greater anxiety than men during the COVID-19 pandemic. In both women and men, diet quality showed a significant inverse association with anxiety; the strength of the association did not differ by sex, and by-sex differences in overall diet quality were not detected.

Though total MedDiet scores did not differ significantly between women and men, several individual diet score components did differ by sex. For example, women reported consuming significantly more daily servings of vegetables than men, a finding consistent with reports in the US [25]. Daily servings of vegetables were weakly but significantly inversely associated with anxiety in women, but not in men. In the present survey, women reported using olive oil as their primary culinary oil significantly more frequently than men. This finding diverges from a report of the Spanish participants in the European Prospective Investigation on Cancer (EPIC) cohort, in which women consumed less olive oil than men [26]. Nevertheless, consuming olive oil was inversely associated with anxiety in women, but not in men. On the other hand, women also reported consuming significantly more weekly servings of sweet baked goods than men. While this finding fits well with the concept of emotional eating, whereby individuals overconsume unhealthy foods such as sweets in response to negative emotions such as anxiety [27], an association between the number of weekly servings of sweet baked goods and the GAD-7 score was not observed in the present survey in women or men. Men reported consuming significantly more sweetened/carbonated drinks than women. In both women and men, consumption of sweetened/carbonated drinks was positively associated with anxiety, and the strength of the association did not differ by sex. This association was also observed regarding red/processed meat intake. Men reported consuming more weekly servings of these foods, but red/processed meat intake was positively associated with anxiety in both women and men; a significant by-sex difference in the strength of this association was not detected. Though men reported consuming more alcohol, a positive association between alcohol consumption and anxiety was observed only in women, and a significant difference in the strength of this association by sex was observed. Compared to women, men reported consuming more weekly servings of legumes; however, no association between legume intake and anxiety was observed in men, and a positive association was detected in women. Weekly servings of hummus/tahini were inversely associated with anxiety in both women and men, with no between-sex difference in the strength of the association, though men reported higher intake. Women and men did not differ in their reported intake of weekly savory baked goods or salty snacks. Consumption of these foods was more strongly associated with anxiety than other items in the MedDiet score. Interestingly, a food intake report during COVID-19 lockdowns indicates that almost a quarter of Italian respondents increased their intake of salty snacks during the pandemic [28].

The impact of the pandemic has been particularly difficult for women, according to several investigators. In a commentary on lifestyle and stress management during COVID-19, it was suggested that women were more likely to suffer from stress and depression, and that this would be accompanied by such unhealthy behaviors as unhealthy diet, smoking, reduced physical activity and increased alcohol use [29]. In the present study, more women than men reported a worsening of diet quality since the onset of the pandemic, though MedDiet scores did not differ by sex. Alcohol use was greater in men than women, but a change in alcohol use since the onset of the pandemic was not queried.

When anxiety was categorized, women reported mild (GAD-7 ≥ 5) and moderate to severe (GAD-7 ≥ 10) anxiety significantly more frequently than men. This rate is considerably higher than the prevalence of any anxiety in the general population, which is estimated to be 1.9–5.1% [21]. In a study of sleep quality and anxiety during the COVID-19 pandemic, 80% of women vs. 50% of men reported increased general anxiety, and women had significantly greater anxiety scores; however, these differences did not translate into differences in sleep quality [30]. Consistent with this, women reported more clinical symptoms than men in an Australian study of anxiety and depression during the initial phase of the COVID-19 pandemic [31]. In our model of moderate to severe anxiety, female sex more than doubled the odds of this endpoint. Additionally, every one-point increase in the MedDiet score was associated with a 7% decrease in the odds of moderate to severe anxiety, and participants who reported that their pre-pandemic diets were healthier than their current diets had a 74% increase in odds of moderate to severe anxiety. These finding are consistent with those of a study of associations between anxiety and dietary habits conducted in Latin America during COVID. In that study, “dysfunctional” food intake was associated with increased anxiety, and women reported greater disruption to their dietary patterns including more emotional eating [32].

Several mechanisms have been suggested to explain associations between diet and anxiety. For example, consumption of a high-fat diet appears to increase levels of corticosterone and circulating inflammatory cytokines, disrupting intracellular cascades involved in synaptic plasticity, glucose homeostasis [33]. Diets high in refined carbohydrates increase the release of pro-inflammatory cytokines in microglial cells, increasing nitric oxide production, which is associated with anxiety in humans [34].

Inflammation has been identified as an explanatory mechanism underlying both cardiovascular risk and depression, and this association may be modified by sex. It has been suggested that under stress, women respond with increased intake of unhealthy food, reduced physical activity, and poor sleep quality, leading to further increased stress. This constellation increases risk for both cardiovascular disease and depression in women [35].

Age was associated with a reduced risk of moderate to severe anxiety among survey participants. This is consistent with previous reports indicating a decline in anxiety beginning in middle age and continuing through old age [36]. Female sex increased the risk of anxiety, consistent with the literature.

It is interesting that responding to the survey in English (vs. any other language) increased the odds of moderate to severe anxiety by more than four-fold. The language in which participants completed the survey may reflect their culture; however, the survey did not query whether the language in which the survey was completed was native to respondents or the dominant language of their residence. Anxiety has been shown to differ across cultures [37], and the by-language differences in anxiety could reflect true cross-cultural differences. Alternatively, the by-language differences in anxiety could represent differences in the severity of the pandemic and/or public health messaging in locations where surveys were answered in English. Individuals who responded in English had a median (IQR) age of 29 (11) years compared to 32 (21) years in those who completed the survey in other languages, and age was inversely associated with anxiety.

As in all studies, these findings must be considered in the framework of design limitations. The data are cross-sectional, so the causality of associations between sex, diet quality, and anxiety cannot be ascertained. Additionally, many of the observed significant correlations were relatively weak, likely due to the large sample size. Further, the sampling method was a convenience sample, bringing into question the external validity of the findings. Participation was predicated on using social media. Participant characteristics, including relatively young age, largely female, and highly educated, suggest that the study sample is different from the general world population in many respects. Nevertheless, the present survey presents findings of a very large, multi-national group of respondents during the rapid contagion period and lockdowns of the early COVID-19 pandemic.

Another potential limitation of the study is that the various countries reported by participants as their current location were not undergoing identical levels of lockdown or other mitigation/prevention measures at the time of the survey. Thus, the anxiety levels reported undoubtedly reflect the increased anxiety experienced generally as a result of the pandemic, and may have been further heightened in some locations by public health measures.

In conclusion, in this very large, international survey, women reported higher anxiety scores than men regardless of the language in which the survey was completed. Diet quality did not differ by sex and was inversely associated with anxiety in both women and men. Despite a lack of by-sex differences in overall diet quality as measured by the MedDiet score, some individual score components did differ by sex, and their associations with anxiety also often differed by sex. The examination of score components in addition to total score deserves further research. If a similar median score misses underlying differences in dietary intake patterns, then components of diet quality scores should be examined when comparing between groups. These dissimilarities may explain some of the variability in between-group morbidity and mortality observed between women and men. In the future, by-sex differences should be considered when assessing diet-disease association, and overall scores should be decomposed to components. Further, the findings of the present study suggest that preventive and wellness public health campaigns during health-related events such as pandemics should be sex-specific. The increase in anxiety and deterioration of diet during the early period of the COVID-19 pandemic in women implies a need to address these issues for women in particular.

## Figures and Tables

**Figure 1 nutrients-13-04193-f001:**
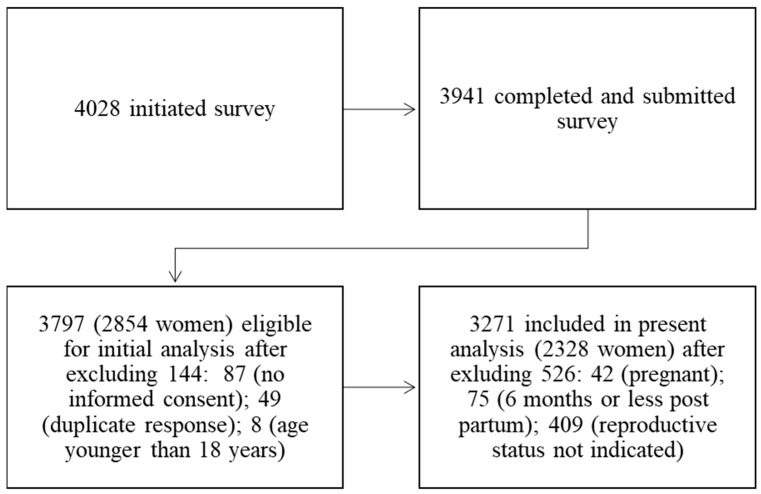
Participant Dispensation.

**Figure 2 nutrients-13-04193-f002:**
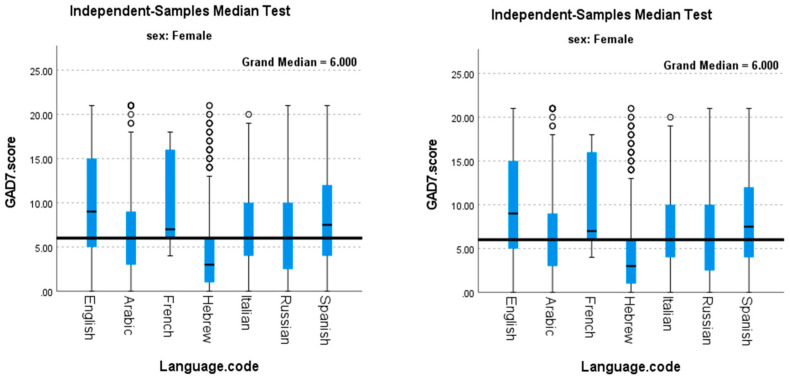
GAD-7 Scores by Sex and Language in which Survey was Completed.

**Table 1 nutrients-13-04193-t001:** Study Population Characteristics by sex.

Characteristic	Women (*n* = 2328)	Men (*n* = 943)	*p*-Value
Age (years, median (interquartile range))	30 (16)	31 (19)	0.079
Language n (%)			0.005
Arabic	203 (8.7)	68 (7.2)	
English	1079 (46.3)	418 (44.3)	
French	14 (0.6)	17 (1.8)	
Hebrew	874 (37.5)	390 (41.4)	
Italian	81 (3.5)	29 (3.1)	
Russian	35 (1.5)	8 (0.8)	
Spanish	42 (1.8)	13 (1.4)	
Health Status n (%)			0.81
No COVID-19	2259 (97.1)	919 (97.5)	
COVID-19 (current or recovered)	13 (0.6)	4 (0.4)	
Suspected COVID-19	55 (2.4)	20 (2.1)	
Present smoker n (%)	297 (12.8)	176 (18.7)	<0.001
Usual Place of employment n (%)			<0.001
Large/medium private sector company	487 (20.9)	254 (26.9)	
Small private sector company	327 (14.0)	132 (14.0)	
Public sector/non-profit	733 (31.5)	232 (24.6)	
Unemployed/retired	523 (22.5)	196 (20.8)	
Independent business owner/freelancer	243 (10.4)	125 (13.3)	
Other	15 (0.6)	4 (0.4)	
Work Status During Pandemic n (%)			0.061
Leave of absence with pay	158 (6.8)	51 (5.4)	
Leave of absence without pay	396 (17.0)	154 (16.3)	
Retired	61 (2.6)	28 (3.0)	
Still Working	1144 (49.1)	500 (53.0)	
Unemployed	447 (19.2)	149 (15.8)	
Other	122 (5.2)	61 (6.5)	
Education n (%)			<0.001
Fewer than 12 years	28 (1.2)	22 (2.3)	
High school diploma/matriculation	564 (24.2)	289 (30.6)	
Professional license (technician, tradesperson, etc.)	174 (7.5)	85 (9.0)	
Bachelor’s degree	875 (37.6)	314 (33.3)	
Master’s degree or higher	673 (28.9)	228 (24.2)	
Other	14 (0.6)	5 (0.5)	

**Table 2 nutrients-13-04193-t002:** Lifestyle and Dietary Characteristics of the Study Population.

Characteristic	Women (*n* = 2328)	Men (*n* = 943)	*p*-Value
MedDiet Score (median (interquartile range))	9 (3)	9 (3)	0.885
MedDiet Component Questions			
Uses Olive Oil as Main Culinary Fat n (%)	1547 (66.5)	571 (60.6)	0.001
Eats poultry/white meat more than red meat n (%)	1532 (65.8)	549 (58.2)	<0.001
Vegetable servings/day (median (interquartile range))	3 (3)	3 (4)	0.008
Fruit servings/day (median (interquartile range))	1 (2)	2 (2)	0.319
Butter/margarine/cream servings/day (median (interquartile range))	1 (2)	1 (2)	0.229
Sweetened beverages/day (median (interquartile range))	0 (1)	0 (1)	<0.001
Whole grain servings/day (median (interquartile range))	2 (3)	2 (3)	0.262
Unsweetened dairy servings/day (median (interquartile range))	2 (2)	2 (3)	0.826
Red/processed meat servings/week (median (interquartile range))	1 (3)	2 (4)	<0.001
Alcoholic beverages/week (median (interquartile range))	0 (3)	1 (3)	0.002
Legume servings/week (median (interquartile range))	2 (3)	2 (4)	0.042
Fish servings/week (median (interquartile range))	1 (2)	1 (2)	0.095
Nut servings/week (median (interquartile range))	1 (3)	1 (3)	0.979
Hummus/tahina servings/week (median (interquartile range))	1 (3)	1 (4)	<0.001
Sweet baked goods servings/week (median (interquartile range))	3 (5)	2 (4)	<0.001
Savory baked goods servings/week (median (interquartile range))	0 (2)	0 (2)	0.061
Salty snacks servings/week (median (interquartile range))	1 (3)	1 (3)	0.535
Change in diet quality n (%)			<0.001
Healthier prior to pandemic	1459 (63.0)	514 (55.0)	
No difference	559 (24.1)	294 (31.5)	
Healthier after the pandemic onset	297 (12.8)	126 (29.8)	
No response	13 (0.6)	9 (0.95)	
Weight change since the start of coronavirus pandemic n (%)			0.041
Yes, weight gain	585 (25.1)	202 (21.4)	
Yes, weight loss	402 (17.3)	174 (18.4)	
No	735 (31.6)	335 (35.7)	
Don’t know	605 (26.0)	231 (24.5)	
Quantity of weight change among those reporting change			0.102
Weight gained (median (interquartile range))	2 (1.5)	2.3 (2.0)	
Weight lost (median (interquartile range))	−2.3 (1.5)	2.3 (2.0)	
Vegan/Vegetarian n (%)	356 (15.3)	78 (8.3)	<0.001
Takes nutrition supplements n (%)	833 (35.9)	278 (29.5)	<0.001
Minutes of exercise/week prior to pandemic (median (interquartile range))	120 (180)	120 (220)	0.171
Minutes of exercise/week in the past week (median (min-max))	60 (175)	60 (147)	0.27

The Mediterranean Diet Score was calculated such that a given question received a value of 1 if the criterion was met and 0 if it was not; thus that the total score can range from 0 to 17 points.

**Table 3 nutrients-13-04193-t003:** Anxiety Measures.

Characteristic	Women (*n* = 2328)	Men (*n* = 943)	*p*-Value
Total GAD-7 Score (median (interquartile range))	6 (8)	3 (6)	<0.001
GAD-7 score items (median (interquartile range)) *			
Feeling nervous, anxious or on edge	1 (1)	1 (1)	<0.001
Not being able to stop or control worrying	1 (1)	0 (1)	<0.001
Worrying too much about different things	1 (2)	1 (1)	<0.001
Trouble relaxing	1 (2)	0 (1)	<0.001
Being so restless that it is hard to sit still	0 (1)	0 (1)	<0.001
Becoming easily annoyed or irritable	1 (2)	1 (1)	<0.001
Feeling afraid as if something terrible might happen	1 (1)	0 (1)	<0.001

* The GAD-7 scale asks the respondent to refer to the two weeks prior to the survey. Each item on the scale can receive is scored as follows: 0 (not at all), 1 (several days), 2 (more than half of the days), 3 (nearly every day); thus, the total score can receive a value from 0–21.

**Table 4 nutrients-13-04193-t004:** Associations between GAD-7 score and Mediterranean Diet score.

Characteristic	Women (*n* = 2328)	Men (*n* = 943)	*p*-Value *
Total GAD-7 Score and Total Mediterranean Diet Score (rho, *p*)	−0.17, <0.001	−0.15, <0.001	0.76
Uses Olive Oil as Main Culinary Fat (median (IQR))	−0.04, 0.042	−0.04, 0.200	1.0
Eats poultry/white meat more than red meat (median (IQR))	−0.03, 0.076	−0.004, 0,895	0.50
Vegetable servings/day (rho, *p*)	−0.04, 0.032	0.01, 0.873	0.19
Fruit servings/day (rho, *p*)	−0.03, 0.13	−0.02, 0.617	<0.001
Butter/margarine/cream servings/day (rho, *p*)	0.20, <0.001	0.12, <0.001	0.03
Sweetened beverages/day (rho, *p*)	0.11, <0.001	0.16, <0.001	0.19
Whole grain servings/day (rho, *p*)	0.05, 0.032	0.03, 0.353	0.60
Unsweetened dairy servings/day (rho, *p*)	−0.07, 0.001	0.01, 0.728	0.04
Red/processed meat servings/week (rho, *p*)	0.12, <0.001	0.08, 0.016	0.29
Alcoholic beverages/week (rho, *p*)	0.10, <0.001	0.02, 0.652	0.04
Legume servings/week (rho, *p*)	0.052, 0.013	−0.02, 0.500	0.07
Fish servings/week (rho, *p*)	−0.13, <0.001	−0.10, 0.002	0.21
Nut servings/week (rho, *p*)	−0.05, 0.028	−0.08, 0.020	0.44
Hummus/tahina servings/week (rho, *p*)	−0.10, <0.001	−0.16, <0.001	0.11
Sweet baked goods servings/week (rho, *p*)	0.025, 0.224	0.04, 0.260	0.69
Savory baked goods servings/week (rho, *p*)	0.21, <0.001	0.16, <0.001	0.18
Salty snacks servings/week(rho, *p*)	0.29, <0.001	0.29, <0.001	1.0

* *p*-value is for the by-sex comparison of the correlation coefficient rho.

**Table 5 nutrients-13-04193-t005:** Multivariable Logistic Regression Model of Moderate Anxiety or Greater (GAD-7 score ≥ 10).

Variable	Odds Ratio	95% Confidence Interval	*p*-Value
Age (years)	0.988	0.980–0.996	0.002
Sex (Female = 1)	2.31	1.88–2.85	<0.001
MedDiet Score	0.93	0.89–0.96	<0.001
Change in diet (diet deteriorated = 1)	1.74	1.32–2.29	<0.001
Language (English = 1)	4.74	3.94–5.69	<0.001
Employment status (unemployed = 1)	1.36	1.10–1.68	0.004
Constant	0.133		0.029

Female sex (vs. male) was the indicator variables for sex; diet healthier prior to the COVID-19 outbreak (subjective report) was the indicator variable for change in diet (vs. no change or improvement in diet); English was the indicator variable for language (vs. any other language); Unemployed (vs. any other employment status) was the indicator variable for employment status.

## Data Availability

The study is registered at Clinicaltrials.gov (accessed on 16 December 2020), NCT04353934.

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
