# Peer review of "Dietary Changes and Anxiety during the Coronavirus Pandemic: Differences between the Sexes"

_nutrients, 2021, doi:10.3390/nu13124193_

Round 1
Reviewer 1 Report
The manuscript entitled ‘Dietary changes and anxiety during the coronavirus pandemic: by-sex differences’ presents interesting issue, however some corrections are needed
- The abstract should be a single paragraph and should follow the style of structured abstracts, but without headings (according to the Instructions for Authors)
- In this section Authors presented the information associated with the Anxiety during COVID-19 pandemic. This section should be presented – what do we know and what is the background for this study. Some detailed information about other studies associated with emotional eating during COVID-19 are necessary (international context – the situation in other countries should be presented, e.g. https://www.mdpi.com/2072-6643/12/12/3889; https://www.ncbi.nlm.nih.gov/pmc/articles/PMC7535346/). The good background should present the history of problem, the current knowledge and scientific "gap", and then authors should present how their study could fill this gap to justify the study.
- More details must be presented in case of the methodology. The results must be reproducible – so the detailed information is needed
- Please provide the information about validation of the translated questionnaire
- More information is needed about the validity and reliability of each measure. Additionally, any limitations in reliability and validity need to be addressed in the discussion.
- 2. Ethics – Please add the information about number of ethics commission approval (specific reference)
- ‘previously have corona’ – please used proper nomenclature.
- Authors should in their discussion include 3 areas: (1) compare gathered data with the results by other authors, (2) formulate implications of the results of their study and studies by other authors, (3) formulate the future areas which should be studied.
- In discussion more references must be added (compare gathered data with the results by other author!)
- Authors should expanded the limitations of their study (please add the discussion assorted with the limitations) .
- ‘This deserves further research.’ - some sentences are too short
- Figure 2. – maybe authors should put this figure in supplementary materials
Reviewer 2 Report
This is a nice manuscript that examines by-sex differences in diet quality and associations between diet quality and anxiety levels during the COVID-19 pandemic.
The manuscript is not very original however it deals with a very important topic:
I have some comments:
The first concern is related to patient’s selection. It is not clear how the population was selected and how the questionnaire was distributed. If the link has been distributed in different countries it is necessary to identify which ones because the duration of the lockdown and the rigidity of the rules was different in the different countries.
A second concern is related to women perception of stress and depression and the influence of such parameters on lifestyle. Women are more aware of the positive effects of a healthy lifestyle but nevertheless are less adherent to the healthy lifestyle. The reasons are mostly socio-economic, because healthy lifestyles are expensive and time-consuming and women have a lot of time occupied by social roles in the family.
You may find this reference useful:“Mattioli AV, Sciomer S, Maffei S, Gallina S. Lifestyle and Stress Management in Women During COVID-19 Pandemic: Impact on Cardiovascular Risk Burden. Am J Lifestyle Med 2021, 15(3), pp. 356–359 doi: 10.1177/1559827620981014”
Bucciarelli V, Nasi M, et al. Depression pandemic and cardiovascular risk in the COVID-19 era and long COVID syndrome: Gender makes a difference. Trends Cardiovasc Med. 2021 Oct 5:S1050-1738(21)00115-8. doi: 10.1016/j.tcm.2021.09.009. Epub ahead of print. PMID: 34619336; PMCID: PMC8490128.
A comment on this would improve discussion section.
Several articles have been published on lifestyle and diet changes in women during lockdown, I believe that increasing the bibliography would be useful for the reader and would make the manuscript more interesting.
Finally, do the authors believe that their conclusions can be translated into social advertising campaigns specifically dedicated to women? a comment on this could improve the manuscript.
Round 2
Reviewer 1 Report
I appreciate the great efforts that the authors have made in response to my questions and concerns. However, there are some typos in text that should be corrected.